# The First Whole Genome Sequencing of *Agaricus bitorquis* and Its Metabolite Profiling

**DOI:** 10.3390/jof9040485

**Published:** 2023-04-18

**Authors:** Chunhua Zhao, Xi-long Feng, Zhen-xin Wang, Jianzhao Qi

**Affiliations:** 1Hubei Key Laboratory of Natural Medicinal Chemistry and Resource Evaluation, School of Pharmacy, Tongji Medical College, Huazhong University of Science and Technology, Wuhan 430030, China; 2Shaanxi Key Laboratory of Natural Products & Chemical Biology, College of Chemistry & Pharmacy, Northwest A&F University, Yangling, Xianyang 712100, China

**Keywords:** *Agaricus bitorquis*, comparative genome, mitogenome, edible mushroom

## Abstract

*Agaricus bitorquis*, an emerging wild mushroom with remarkable biological activities and a distinctive oversized mushroom shape, has gained increasing attention in recent years. Despite its status as an important resource of wild edible fungi, knowledge about this mushroom is still limited. In this study, we used the Illumina NovaSeq and Nanopore PromethION platforms to sequence, de novo assemble, and annotate the whole genome and mitochondrial genome (mitogenome) of the *A. bitorquis* strain BH01 isolated from Bosten Lake, Xinjiang Province, China. Using the genome-based biological information, we identified candidate genes associated with mating type and carbohydrate-active enzymes in *A. bitorquis*. Cluster analysis based on P450 of basidiomycetes revealed the types of P450 members of *A. bitorquis*. Comparative genomic, mitogenomic, and phylogenetic analyses were also performed, revealing interspecific differences and evolutionary features of *A. bitorquis* and *A. bisporus*. In addition, the molecular network of metabolites was investigated, highlighting differences in the chemical composition and content of the fruiting bodies of *A. bitorquis* and *A. bisporus*. The genome sequencing provides a comprehensive understanding and knowledge of *A. bitorquis* and the genus *Agaricus* mushrooms. This work provides valuable insights into the potential for artificial cultivation and molecular breeding of *A. bitorquis*, which will facilitate the development of *A. bitorquis* in the field of edible mushrooms and functional food manufacture.

## 1. Introduction

Mushrooms are a diverse group of organisms belonging to the kingdom Fungi and are known for their unique physical characteristics, especially their fruiting bodies, which are visible above ground or on the surface of their host substrate. While many species are consumed as food, others are known for their medicinal or psychoactive properties. Mushrooms have been integral part of human culture and traditional medicine for thousands of years and continue to be studied for their diverse properties. One of the primary ways humans utilize mushrooms is through the domestication and artificial cultivation of wild mushrooms. Among these, *Agaricus bisporus*, known as the button mushroom, stands out as one of the most successful examples of human utilization of mushrooms, having originated in Paris in the 18th century [1,2] and becoming a globally cultivated edible mushroom known as the ‘world mushroom’.

In recent years, another member of the *Agaricus* genus, *Agaricus bitorquis*, has attracted attention due to its antibacterial [3,4], antioxidant [3,4,5], immunomodulatory [6], and anticancer [7,8] activities, as well as its selenium enrichment properties [9]. While *A. bitorquis* is a rare wild edible mushroom distributed in various regions of Europe and the Americas [4,10], it is also found in Xinjiang, Qinghai, Inner Mongolia, and Hebei Province in China [5,11], where it is often referred to as the “Dafei mushroom” due to its large size. The wild fruiting bodies of *A. bitorquis* grow beneath the soil at a depth of about 15 cm and are generally distributed in sheets, rarely exposed to the ground. The Bosten Lake area of Xinjiang is a habitat where the local people highly value *A. bitorquis* and it was initially discovered by sheep. Despite the edible attributes and potential medicinal value of *A. bitorquis*, no large-scale cultivation has been reported; however, artificial cultivation has been implemented with promising results [12,13].

Recently, third-generation sequencing technologies have emerged, allowing for more accurate and comprehensive genome sequencing of medicinal and edible mushrooms. These technologies have facilitated research on various aspects of these fungi, including their life cycles, mating types, nutritional patterns, and biosynthesis of bioactive metabolites. As a result, the genomes of several valuable medicinal fungi, such as *Inonotus obliquus* [14], *Hericium erinaceus* [15], *Laetiporus sulphureus* [16], *Antrodia camphorate* [17], and *Inonotus hispidus* [18], have been successfully deciphered, which further aid in their medicinal utilization and industrial development. Moreover, the genome sequencing of some precious wild edible fungi, including *Oudemansiella raphanipes* [19], *Naematelia aurantialba* [20], and *Pleurotus giganteus* [21] have the potential to advance artificial cultivation and strain selection of these species.

The genus *Agaricus* contains a wide range of species with medical and culinary value, including *A. bitorquis* and *A. bisporus*. Despite their shared taxonomic classification, the two species exhibit differences in shape preference and morphology (Figure 1). Furthermore, while *A. bisporus* has been extensively studied [22,23], relatively little is known about *A. bitorquis*, despite its potential as a valuable source of bioactive compounds. To address this gap in knowledge, the whole genome of a wild strain of *A. bitorquis* was sequenced and analyzed. This effort represents the first comprehensive investigation of *A. bitorquis* at the chromosomal level, revealing novel insights into its mating system and carbohydrate metabolic capacity. By comparing the genomes, mitochondrial genomes, and metabolic profiles of *A. bitorquis* and *A. bisporus*, this study provides valuable new information on the genetics and physiology of these important macrofungi. Overall, this work enhances our understanding of the genome of medicinal and edible fungi and represents a significant step forward in the study of the genus *Agaricus*.

## 2. Materials and Methods

### 2.1. Fungal Strain and Strain Culture

The fruiting bodies in their natural habitat (Figure 1A,B) were gathered from wetlands bordering Bosten Lake in the Bayingol Mongolian Autonomous Prefecture, located in Xinjiang Province, China. The sample was identified by examination of zygote morphological characteristics and ITS sequence alignment (Appendix A) of the mycelium, and this analysis revealed the sample to be *A. bitorquis*, subsequently classified as *A. bitorquis* BH01. Tissue isolation was carried out using fresh wild fruiting bodies of the strain BH01. To obtain culturable mycelium, surface-sterilized small pieces of fruiting bodies were cultivated on potato dextrose agar (PDA) plates for 3–4 days. The identified mycelium was then deposited in the Shaanxi Key Laboratory of Natural Products and Chemical Biology, College of Chemistry and Pharmacy, Northwest A&F University. This process was necessary to ensure that the mycelium could be used for further research.

### 2.2. Genome Sequencing, De Novo Assembly, and Annotation

#### 2.2.1. Extraction of Genome DNA

Cultivation of *A. bitorquis* BH01 mycelium was carried out in a controlled environment using potato dextrose broth (PDB) medium at 25 °C with agitation at 200 rpm for one week. The aim was to obtain an adequate amount of fresh and viable mycelia. To ensure the purity and freshness of the mycelium, it was then subjected to a series of washing steps, including centrifugation followed by rinsing with sterile water and centrifugation again to remove excess water. Genomic DNA was extracted from the mycelium using the sodium dodecyl sulphate technique, which involved grinding the mycelium with liquid nitrogen and testing the integrity of the DNA using agarose gel electrophoresis. The specific isolation and purification method is described in a previous document [24]. 

#### 2.2.2. Sequencing and De Novo Assembly

The first step in the sequencing of the *A. bitorquis* BH01 genome involved end repair, addition of A-tails and sequencing of junctions, followed by purification and PCR amplification of the genomic DNA. Prior to library generation, high quality bulk DNA was collected and assessed for purity, concentration, and integrity. To ensure the quality of the libraries, quantification and quality checks were performed using Qubit 2.0. Sequencing was performed on both the Oxford Nanopore PromethION sequencing platform and the Illumina NovaSeq platform using 20 kb and 350 bp insert sizes, respectively. The NECAT (https://github.com/xiaochuanle/NECAT (accessed on 21 April 2021)) tool was used to correct genomic errors and splicing was performed to obtain initial splicing results from the third-generation sequencing data. Two rounds of error correction were then performed using Racon v1.4.7 (https://github.com/isovic/racon (accessed on 21 April 2021)), followed by two rounds of Pilon. Finally, the assembly result was determined after error correction and heterozygosity elimination. 

#### 2.2.3. Gene Prediction and Annotation

BRAKER v2.1.4 (https://github.com/Gaius-Augustus/BRAKER (accessed on 22 April 2021)) was primarily used to predict gene sequences. GeneMark-EX was then used to train the model and AUGUSTUS (https://github.com/Gaius-Augustus/Augustus (accessed on 22 April 2021)) to predict open reading frames (ORFs). INFERNAL v1.1.2 (https://github.com/EddyRivasLab/infernal (accessed on 22 April 2021)) was used to predict and categorize non-coding RNA based on the Rfam database. In addition, RepeatModeler v1.0.4 (https://github.com/Dfam-consortium/RepeatModeler (accessed on 22 April 2021)) was used to generate a repeat library after integration of the Rebase library, while RepeatMasker v4.0.5 (https://github.com/rmhubley/RepeatMasker (accessed on 22 April 2021)) was used to annotate repetitive genomic sequences. Finally, BLAST searches of non-redundant protein sequences from the NCBI, Swiss-Prot, COG, and KEGG databases were performed to annotate the gene products. 

### 2.3. Comparative Genomics Analysis

To analyze and visualize genome collinearity, McscanX (https://github.com/wyp1125/MCScanX (accessed on 4 March 2023)) was employed. Single-copy genes were utilized to undertake comparative genomic analysis within *Agaricus* species, and the results were visualized using jVenn (http://jvenn.toulouse.inra.fr/app/index.html (accessed on 2 March 2023)). To calculate the synonymous substitution rate (Ks) between two *A. bisporus* species and *A. bitorquis* BH01, wgd v1.1 (https://github.com/arzwa/wgd (accessed on 13 March 2023)) was employed. a genome-wide duplication analysis was performed. To acquire collinear block pairs among the species, wgd was utilized with coding-gene sequences, amino acid sequences, and genome annotation files. Subsequently, ParaAT v2.0 was employed to convert the homologous protein sequence pairs to CDS pairs. Homologous sequence pairings were estimated by wgd, and the results were displayed using Rstudio v4.20.

### 2.4. Phylogenomic Analysis

Phylogenetic analysis was performed to investigate the evolutionary relationships between *Agaricus* strains and 27 other representative strains of Basidiomycetes. Single-copy homologous genes were identified using OrthoFinder v2.5.4 with the parameters “-S diamond -M msa -T raxml-ng”. A total of 344 single-copy orthologous sequences from 24 strains were used to predict divergence time using the MCMC tree method (http://abacus.gene.ucl.ac.uk/software/paml.html (accessed on 12 March 2023)). The calibrated points of several groups of recent ancestor divergence times were queried in timetree.org (http://www.timetree.org/ (accessed on 11 March 2023)), including *Hericium alpestre* vs. *Stereum hirsutum* (91.8–195.5 MYA), and *Ganoderma sinense* vs. *Laetiporus sulphureus* (99–152.5 MYA). 

### 2.5. CAZy Family and Cytochrome P450 Analyses

To annotate and classify the genes encoding carbohydrate-active enzymes (CAZymes) from the genomes of *A. bitorquis* BH01 and other white rot fungi, the CAZy database (http://bcb.unl.edu/dbCAN2/ (accessed on 8 March 2023)) was used with the HMMER 3.2.1 tool (filter parameter E-value < e^−5^; coverage > 0.35). A bubble plot of the CAZyme analysis was generated for *I. hispidus* using the Complex Heatmap package in Rstudio v4.20. 

The Diamond 2.9.0 (e-value > e^−5^) was used with the Hmmer package to predict P450s and annotate the target protein sequence. Reference P450 sequences for cluster analyses were obtained from the Fungal Cytochrome P450 database (http://p450.riceblast.snu.ac.kr/index.php?a=view (accessed on 8 March 2023)). For the phylogenetic tree analysis, 115 predicted P450 proteins from *A. bitorquis* BH01 and several other Basidiomycetes selected from the Fungal P450 database were clustered with accurate classification. A maximum likelihood tree was constructed using IQ-tree 2.2.3 (https://github.com/iqtree/iqtree2 (accessed on 10 March 2023)) with the options “-m MFP -bb 1000 -alrt 1000 -abayes -nt AUTO”.

### 2.6. Sequencing, Assembly, Annotation, and Comparative Analysis of Mitogenome

The mitochondrial genome sequencing was conducted utilizing a hybrid approach of Illumina NovaSeq and Nanopore sequencing technology, generating an average genome coverage of 150-fold with a paired-end library. The de novo assembly of the reads was accomplished using SOAPdenovo 2.0425 (https://github.com/aquaskyline/SOAPdenovo2 (accessed on 25 April 2021)). The sequencing data derived from the Nanopore platform were corrected by aligning with the Illumina sequencing reads utilizing BLASR26 (https://github.com/jcombs1/blasr (accessed on 25 April 2021)), and then assembled using the Celera Assembler (https://www.cbcb.umd.edu/software/celera-assembler (accessed on 25 April 2021)). The accuracy of the sequencing reads was further improved by correcting them again based on the Illumina data, following the generation of reliable scaffolds. The final output represents the complete genome sequences with high accuracy and reliability. The mitogenomes of *I. Hispidus* and *P. gilvus* were de novo assembled from Nanopore raw reads using minimap2 v2.17-r94 (https://github.com/lh3/minimap2 (accessed on 21 February 2023)) and miniasm v0.3-r179 (https://github.com/lh3/miniasm (accessed on 21 February 2023)), and further refined using racon v1.4.20 (https://github.com/isovic/racon (accessed on 21 February 2023)) and pilon v1.23 (https://github.com/broadinstitute/pilon (accessed on 21 February 2023)), based on Illumina data. The final assemblies were assessed for quality using samtools (http://www.htslib.org/ (accessed on 21 February 2023)). 

The mitogenomes were subjected to annotation using the website MFannot (https://megasun.bch.umontreal.ca/apps/mfannot/ (accessed on 25 March 2023)), utilizing the genetic code 4 to predict protein-coding genes (PCGs), tRNA genes, rRNA genes, and partial open reading frames. The annotation process involved manual proofreading, and the tRNA and rRNA genes were further validated using RNAweasel (https://github.com/BFL-lab/RNAweasel (accessed on 27 February 2023)) and tRNAScan (https://www.psc.edu/resources/software/trnascan-se/ (accessed on 27 February 2023)), respectively. The type I intron was also evaluated for its adherence to normal sequence characteristics using RNAweasel. MAFFT (https://mafft.cbrc.jp/alignment/software/ (accessed on 28 February 2023)) and blast analysis (https://blast.ncbi.nlm.nih.gov/Blast.cgi (accessed on 28 February 2023)) were employed to confirm the starting and ending positions of *rns*, *rps3*, the 14 conserved PCGs, and intron insertion sites. Additionally, ORF Finder (https://www.ncbi.nlm.nih.gov/orffinder/ (accessed on 1 March 2023)) was utilized to explore open reading frames in intergenic regions and intron regions exceeding 300 bp, while Blastn and Blastp were employed to ascertain the starting points and functions of ORFs within the intron. Finally, OGDraw v1.2 (https://chlorobox.mpimp-golm.mpg.de/OGDraw.html (accessed on 1 March 2023)) was utilized to create graphical maps of the complete mitogenomes.

### 2.7. Metabolite Profiling Comparison

Twenty grams of fresh substrates of *A. bitorquis* and *A. bisporus*, respectively, were extracted with ethyl acetate, concentrated, and then used for mass spectrometric (MS) quantification. The High-Resolution MS detection was carried out using AB Sciex TripleTOF 6600 mass spectrometer in both positive-ion and negative-ion modes. Molecular network analysis of HPLC-HRMS data of crude extract was performed using GNPS (https://gnps.ucsd.edu (accessed on 17 March 2023)) with default parameters. The molecular network was visualized finally by Cytoscape 3.9.1.

### 2.8. Data Availability

The ITS sequence of strain *A. bitorquis* BH01 has been deposited in the National Center for Biotechnology Information (NCBI) GenBank under accession number OQ581725. The final genome assembly results and related data of *A. bitorquis* BH01 have been submitted to the NCBI under the BioProject PRJNA946023 and BioSample SAMN33801832, respectively. The mitogenome sequence and annotation of strain *A. bitorquis* BH01 has been deposited in the NCBI GenBank under accession number OQ571893. 

## 3. Results

### 3.1. Genome Sequence Assembly and Annotation of A. bitorquis BH01

A total of 34,235,978 clean reads were generated, resulting in 5,135,396,700 bases with a GC content of 45.19% (Appendix A). These reads were assembled into a high-quality genome of 32.35 Mb, consisting of 22 contigs (Figure 2A, Table 1). The N50 value of the assembly was 1,791,120 bp (Appendix A). The presence of two peaks with a 2-fold relationship in the *K*-mer curve indicated that the genome of *A. bitorquis* BH01 was heterozygous, with a heterozygosity of 0.665% (Appendix A), suggesting that this fungus was a dikaryon. The completeness of the genome assembly was assessed by a coverage of 99.83% (Appendix A) and a BUSCO value of 90.8% (Appendix A) based on the fungi_odb10 database.

There were 10,028 protein-coding genes predicted by BRAKER, equipped with Augustus. These genes had an average length of 1883.52 bp, consisting of 67,319 exons and 67,319 introns, with average lengths of 180.16 bp and 73.58 bp, respectively (Appendix A). In addition, various non-coding RNAs including 109 tRNAs, 25 rRNAs, 11 snRNAs, and 1 sRNA were predicted (Appendix A). Repeat sequence analysis revealed the presence of 11,738 repeats with a total length of 7,694,206 bp, accounting for 23.79% of the whole genome. Among these repeats, four scattered repeats, namely SINE, LINE, LTR, and DNA transposons, accounted for 0.01% (6), 4.73% (1041), 12.16% (2613), and 7.72% (1379), respectively (Appendix A). These findings provide insight into the genome structure of the mushroom under study. 

To achieve a comprehensive functional annotation of protein-coding genes, sequence similarity analysis and motif similarity search were performed on 10,028 genes based on nine public databases (NCBI nr, Pfam, eggCOG, Uniprot, KEGG, GO, Pathway, Refseq, Interproscan) (Appendix A). The annotation results from the Nr library showed that 9253 genes, representing 92.27% of all protein-coding genes, were annotated. Among these genes, 49.67% matched *A. bisporus* var. bisporus H97 and 41.98% matched *A. bisporus* var. bumeti JB137-S8 (Appendix A), indicating a close relationship between A. *bitorquis* and *A. bisporus*. However, 4.99% of the genes matched *Leucaganius* sp. SymC. Cos (Appendix A), reflecting the intergeneric variability within the genus *Agaricus*. 

Functional annotation of protein-coding genes in strain BH01 revealed their functional diversity in different databases. Among the 5021 genes annotated by the GO database, the classification of cellular components was the most prominent group (Appendix A). Based on the COG database, 976 genes were identified, with 154 genes belonging to the J group, which is related to translation, ribosomal structure, and biogenesis (Appendix A). The KEGG database identified 5010 genes involved in five types of pathways, with the highest number of genes involved in metabolic pathways (Appendix A). In addition, a motif search using the Pfam database identified 7023 genes, of which the top 20 motifs with the most annotated genes are shown in Appendix A. These results highlight the functional diversity of protein-coding genes in strain BH01 from different perspectives and levels of annotation.

### 3.2. Identification of the Mating Genes

In the sexual development of mushroom-forming fungi, mating is a crucial step that is guided by specific mating loci. The mating type (MAT) loci are located in different genomic regions [26]. Heterozygous cooperation, which accounts for up to 90% of fungal mating types, could be classified into bipolar and tetrapolar mating types. Among these, the tetrapolar mating system is the most widespread and complex sexual reproduction control system found in Basidiomycetes to date [27,28]. Given the unknown reasons for the formation of the large fruiting bodies and the potential demand for cultivation of *A. bitorquis*, it is necessary to analyze and identify its mating system. 

In this research, the MAT-A locus was found to be located on Chr 2 by homology search with the mitochondrial intermediate peptidase (*mip*) codon gene and a homeodomain transcription factor-codon gene as probes, which are from *A. bisporus* var. bisporus H97 [23]. The MAT-A locus of *A. bitorquis* contains a glycosyltransferase family 8 protein codon gene (*glgen*, g4765), two unknown conserved fungal protein-codon genes (*β*-*fg*, g4764 and g4761), three homeodomain transcription factors-codon genes (*HD1*, g4759, as well as *HD2*, g4763, and g4760), and a *mip* (g4758). The order of the genes present in clusters on the MAT-A locus is consistent with those of *A. bisporus* var. bisporus H97 (Figure 2B), whereas the MAT-B locus contains at least four unclustered *ste3*, including g142, g182, g7283, and g9071. Of these, g142 and g182 are located on ctg14, while g7283 and g9071 are located on chr3 and chr12, respectively (Figure 2B). The current analysis has shown that the mat-A and mat-B loci of *A. bitorquis* are not located in the same contig, indicating the presence of a tetrapolar mating system [27,28]. However, this finding only scratches the surface of the intricate genomic architecture of *A. bitorquis* mating type loci. Further investigation is essential to gain a full understanding of the mechanisms underlying sexual reproduction in *A. bitorquis*, which is crucial to elucidate the evolutionary trajectory of this fascinating fungal species.

### 3.3. Phylogenomic and Evolutionary Analysis 

The present study used a phylogenomic approach based on an alignment of 344 single-copy orthologous genes from 60,935 orthogroups to elucidate the evolutionary relationships among 29 fungal species, including two *Agaricus* species (Figure 3). The inferred phylogenetic tree was strongly supported by bootstrap values. The mean divergence time between Agaricales and Polyporales was estimated to be 169.78 Mya with 95% highest posterior density (HPD) of 115.50–294.61 Mya, while that between these two orders and Hymenochaetes was estimated to be 204.64 Mya (95% HPD of 125.66–348.17 Mya). Furthermore, the emergence of *Agaricus* was estimated to have occurred at the crown age of 64.97 Mya (95% HPD of 41.30–94.06 Mya), with the divergence between *A. bitorquis* and *A. bisporus* estimated at 8.21 Mya (95% HPD of 4.28–12.34 Mya) (Figure 3).

Interestingly, gene family contraction events were found to be more frequent than gene family expansion events in the evolutionary history of the 29 fungal species studied (Figure 3). In particular, *Agaricus* and *Cortinarius* (or *Psilocybe*) showed the most significant contraction events, affecting a total of 313 gene families. In the genus *Agaricus*, 242 and 167 gene families were contracted in *A. bitorquis* and *A. bisporus*, respectively, whereas 438 and 125 gene families were expanded in these two species, respectively. These results suggest that the genus *Agaricus* has undergone considerable changes during its evolutionary history (Figure 3).

### 3.4. CAZymes Analysis 

Edible fungi are widely cultivated on cellulose- and lignin-rich substrates such as wood chips and straw. The ability of edible fungi to degrade and utilize these substrates is closely related to their expression of CAZymes [29,30]. Given the potential for large-scale cultivation of *A. bitorquis* due to its nutritional value, we investigated its CAZyme repertoire. Our analysis identified 114 genes encoding 123 CAZymes, consisting of 66 glycoside hydrolases (GHs), 22 auxiliary activities (Aas), 14 glycoesterases (Ces), 9 carbohydrate-binding modules (CBMs), 8 polysaccharide lyases (PLs), and 4 glycosyltransferases (GTs). Notably, nine CAZyme-encoding genes (g227, g412, g3471, g3563, g3645, g3914, g7640, g8449, and g9072) were found to possess two functional structural domains (Appendix A, and Dataset_1). When the CAZyme profiles of *A. bisporus* BH01 were compared with those of *A. bisporus* var. burnettii H119 (Figure 4A, Appendix A), no significant differences were observed. In addition, we compared the CAZyme repertoires of A. *bisporus* BH01 and 17 other edible fungi and found that the number and types of CAZymes were not species-specific (Figure 4B, Appendix A). The strains *Lactarius deliciosus* EDB83 and *Paxillus involutus* ATCC 200,175 showed the greatest similarity to *A. bisporus* BH01 in terms of their CAZyme profiles, whereas *A. bisporus* var. burnettii H119 did not (Figure 4B).

### 3.5. Cytochrome P450 Family Analysis

Cytochrome P450s (CYPs) are a ubiquitous class of enzymes found in living organisms and play essential roles in various biological processes. In fungi, the CYP51A1 subfamily is typically responsible for the critical enzyme C14α-demethylation, which is involved in the synthesis of ergosterol [31]. Consequently, CYPs have emerged as a promising target for the inhibition of pathogenic fungi. To gain insight into the functional gene composition of *A. bitorquis*, the number and type of its P450s were analyzed and investigated. Using Pfam prediction based on domain features, we screened a total of 100 P450-encoding genes (115 P450 proteins) in the genome of *A. bitorquis* BH01. Using cluster analysis, we identified clear classifications based on the evolutionary association of these 115 protein sequences with representative basidiomycete P450 sequences from the Fungal Cytochrome P450 Database (Appendix A). These clustering results provided a clear indication of the categorization of the P450s of *A. bitorquis* BH01. In addition, the cluster analysis revealed the presence of 11 CYP subfamilies and ten uncertain groups (Figure 5). Among the identified CYP families, CYP5144 had the highest number of members with 38, followed by CYP512 with 13, and the remaining families had less than 10 members. In particular, 29 P450s were scattered in ten uncertain groups, accounting for 25.22% (Figure 5). These unidentified P450s suggest the presence of potential new P450 types that require further analysis and characterization.

### 3.6. Comparative Genomic Analysis within Agaricus Species

The size difference between the fruiting bodies of *A. bitorquis* and *A. bisporus* makes us curious to know how their genomes differ. Comparative genome analysis shows that the genome size of *A. bitorquis* BH01 is a slightly larger than that of *A. bisporus* var. bisporus H97 and burnettii H119, but encodes fewer proteins (Table 1). Moreover, the genome assembly quality of *A. bitorquis* BH01 is superior to those of *A. bisporus* mushrooms, as indicated by the numbers of scaffolds and contigs, as well as Scaffold L50 (Table 1). Collinearity analysis shows that most genomic regions of *A. bitorquis* BH01 are syntenic to those of *A. bisporus*, with chr1-chr9 of BH01 showing high synteny to specific regions of the *A. bisporus* genomes (Figure 2A).

Comparative analysis of orthologous groups between the three *Agaricus* variants identified a total of 6798 groups, with *A. bitorquis* containing more unique orthologous groups (125) than *A. bisporus* var. bisporus H97 (89), but fewer than burnettii H119 (157). The two *A. bisporus* share more orthologous groups (1143) than that of each of them share with *A. bitorquis* BH01 (236 and 616), respectively (Figure 2C). This normal difference suggested that the intraspecific differences of the two most common *A. bisporus* species variants were smaller than the interspecific differences between them and *A. bitorquis*. To gain further insight into these differences, a genome-wide duplication analysis based on synonymous mutation rates (Ks) was performed. The trends of the Ks curves of the two species of *A. bisporus* to *A. bitorquis* were consistent, but the trend of Ks curves within *A. bisporus* were not consistent with it (Figure 2D). This finding reflects the intraspecific uniformity within *A. bisporus*, and interspecific difference between *A. bitorquis* and *A. bisporus* species.

### 3.7. The Mitogenome of A. bitorquis and Comparative Mitogenomic Analysis within Agaricus Species

The mitochondrion, a critical organelle involved in energy production, ageing, and various physiological processes, has received considerable attention for its utility in deciphering fungal phylogeny, evolution, species identification, taxonomic research, variety protection, and fungal breeding [32,33]. In this regard, the mitochondrial genome of *A. bitorquis* was well-assembled and annotated, alongside its chromosomal genome. The circular DNA molecule of the complete mitogenome spans 153,897 bp with 29.01% GC content and harbors 14 conserved protein-coding genes (PCGs) associated with oxidative phosphorylation, an additional PCG (*rps3*), 27 tRNA genes, 2 rRNA genes (*rns* and *rnl*), and 58 un_ORFs of unknown function or structure (Figure 6A, Table 2). 

Further comparative analyses of the mitogenomes of *A. bitorquis* and *A. bisporus* revealed differences and similarities. Despite their different sizes, both genomes have very similar GC contents and intron numbers. The longer mitogenome of *A. bitorquis* carries more un_ORFs but fewer tRNAs, whereas *A. bisporus* shows the opposite pattern (Table 2). In addition, synteny analysis based on nucleic acid sequences revealed several large regions of highly similar sequences between *A. bitorquis* and *A. bisporus* (Figure 6B). Considering the conservation of PCGs and the potential influence of GC content on the reading frame [34], a comparison of the GC content of 15 PCGs in the two species showed little difference between the two *Agaricus* species (Figure 6C, Appendix A).

**Table 2 jof-09-00485-t002:** Mitogenomic comparison between *A. bitorquis* and *A. bisporus*.

Entry	*A. bitorquis* BH01	*A. bisporus* var. *bisporus* H97
Total length (bp)	153,897	135,055
GC content (%)	29.01%	29.07%
AT-skew	0.019	0.011
GC-skew	0.0304	0.0088
No. of tRNA	27	35
No. of un_ORFs	58	29
No. of introns	47	46
Length of the *cox1* gene (bp)	30,178	29,908
References	This study	[35]

### 3.8. Metabolic Profiling Comparison between A. bitorquis and A. bisporus

*Agaricus bisporus*, the best-known member of the *Agaricus* genus, is not only a globally consumed mushroom with high nutritional value, but also because of its significant medicinal value and prospects for complementary and alternative medicine [36,37]. As an emerging member of the genus *Agaricus*, *A. bitorquis* shows great potential in biological activities such as immunomodulation [4], and anticancer [5,6]. Therefore, to examine the differences between the two in terms of bioactive small molecules, a GNPS-based metabolite profile comparison was performed. The visualized molecular network showed that the chemical compositions and contents of the two species are significantly different (Appendix A).

Furthermore, totals of 23 compounds were identified by comparison of their MS and MS2 characteristics with the previous literature, including γ-L-Glutaminyl-3,4-benzoquinone (**1**), blazeispirols B-E (**2**–**5**), (22*E*,24*R*)-3*β*,5*α*,6*β*,9*α*,14*α*,25-hexhydroxyergosta-7,22-diene (**6**), blazeispirols X and Y (**7**–**8**), blazeispirol A (**9**), demethylincisterol A_3_ (**10**), 11,12-dihydroxy-15-drimeneoic acid (**11**), benzoyl-ergostane (**12**), blazeispirols I and F (**13**–**14**), ergosterol (**15**), brefeldin A (**16**), 3β,11,12-trihydroxydrimene (**17**), 5-methyl-tryptophan (**18**), melatonin (**19**), agaritine (**20**), *N*-(γ-L-glutamyl)-4-hydroxyaniline (**21**), β-*N*-(γ-glutamyl)-4-formylphenyl-hydrazine (**22**) (Figure 7 and Appendix A, Table 3). Among the 18 shared compounds, blazeispirols and lanostane triterpenoids were the most abundant, and most of them (**2**–**9**) were in the same network (Figure 7A). Blazeispirols (**2**–**5**, **7**–**9**, and **13**–**14**), a class of des-A-ergostane-type compounds, clustered with lanostane derivatives in the same network, suggesting their natural correlation. In addition, *A. bitorquis* and *A. bisporus* have their own unique metabolites, **18** and **19**, and two identified indole derivatives appear in the unique network of *A. bisporus* (Figure 7B), whereas, **20**–**22**, three identified y-glutamyl-substituted arylhydrazine derivatives appear in the unique network of *A. bitorquis* (Figure 7C). Metabolic profiling revealed similarities and differences in the metabolites of *A. bitorquis* and *A. bisporus*, reflecting the diversity and complexity of metabolites within the genus *Agaricus*.

## 4. Discussion

The genus *Agaricus* is a well-known taxon in the order Agaricaceae, but their genomes have not been extensively investigated. This study presents the first report of the genome of a wild member of the genus *Agaricus*, *A. bitorquis*, at the chromosome level. Recent advances in technology have enabled the assembly and annotation of *A. bitorquis* to be of superior quality to that previously reported for *A. bisporus*. Using single-copy genes to infer an evolutionary tree, the divergence of *A. bitorquis* and *A. bisporus* is estimated to have occurred approximately 8.21 Mya (Figure 3). However, the expansion and contraction of gene families experienced by both species since their divergence has resulted in obvious differences in their genome sizes (Table 1). Comparative genome analysis based on orthologous groups reveals interspecific convergence between *A. bitorquis* and *A. bisporus* and intraspecific variation in *A. bisporus* (Figure 2C). In contrast, Ks-based evolutionary pressure selection analysis explains the interspecific variability between *A. bitorquis* and *A. bisporus*, and the intraspecific uniformity observed within *A. bisporus* (Figure 2D). These results provide valuable insights into the complexity of the genome of the genus *Agaricus*.

The exaggerated fruiting body size of *A. bitorquis* compared to *A. bisporus* is another point of interest (Figure 1). In particular, comparative genomics revealed the presence of 125 orthologous genes unique to *A. bitorquis*, which may hold the key to unravelling this morphological difference. Functional annotation and clustering analysis of the 278 proteins encoded by these 125 orthologs has identified four *A. bitorquis*-specific transcription factor-encoding genes (g3057, 5076, 6604, and 8494) and several other promising candidate genes (Appendix A). However, it remains to be determined whether these orthologs are indeed associated with fruiting body development in *A. bitorquis*, which will require further validation by transcriptome comparisons and other means. To extend this research, we sought to investigate it using genes involved in the development of *A. bisporus*-derived fruiting bodies. Specifically, we used genes that have been implicated in fruiting body development, such as those related to ATP synthase subunits *atpD* and septin protein gene *sepA* [49], urease and tyrosinase encoding genes [50], polyphenoloxidase genes [51], and specifically expressed genes discovered through transcriptomics [52], as probes for genome mining in *A. bisporus*. Our comparison of the homologous genes between *A. bitorquis* and *A. bisporus* revealed important similarities (Appendix A), which serve as a reference for uncovering the underlying mechanisms responsible for the oversized fruiting bodies in *A. bitorquis*.

Despite the general trend of morphological stability within the same genus mushrooms, some exceptions exist, as is the case for *A. bitorquis*. This species exhibits significant variation in size, which is not commonly observed in other members of the genus *Agaricus*. Although bioactive compounds in fungi are highly diverse, they often exhibit species-specific distribution patterns. For instance, Ganoderic acids are tetracyclic triterpenoids that are characteristic metabolites of the genus *Ganoderma* [53], and styrylpyrones are a group of polyphenols compounds commonly found in the genus *Phellinus* and *Inonotus* species [54,55]. Interestingly, the des-A-ergostane-type compounds, blazeispirols, were initially discovered in *Agaricus blazei* [39]. Here, the metabolite profile comparison revealed the presence of blazeispirol compounds in both *A. bitorquis* and *A. bisporus*. This finding suggests that the blazeispirol compounds may be characteristic of the whole *Agaricus* genus. Moreover, the availability of the *A. bitorquis* genome provides a necessary condition to investigate the biosynthesis of these compounds further.

## 5. Conclusions

*A. bitorquis* is a giant wild mushroom with various biological activities. In this study, we present for the first time the whole genome and mitogenome of *A. bitorquis*. The chromosome-level assembly and functional annotation of the genome offer crucial clues for investigating the mating loci and CAZymes of this wild mushroom, which will aid in its artificial cultivation. Cluster analysis reflects the diversity of P450s in *A. bitorquis*. Comparative genomic and mitogenomic analyses reveal distinct genetic compositions among the *Agaricus* genus, while phylogenetic and evolutionary analyses of the genus *Agaricus* reveal contraction and expansion of their gene families, and species divergence times. Additionally, molecular network-based metabolite analysis revealed differences in chemical composition and content in the fruiting bodies of *A. bitorquis* and *A. bisporus*, and suggests that blazeispirols are the characteristic compounds of the genus *Agaricus*. This study not only fills the gap in the genetic information of *A. bitorquis*, but also provides significant insights into the genome of mushrooms in the *Agaricus* genus. These results will pave the way for the development of functional foods using *A. bitorquis*.

## Figures and Tables

**Figure 1 jof-09-00485-f001:**
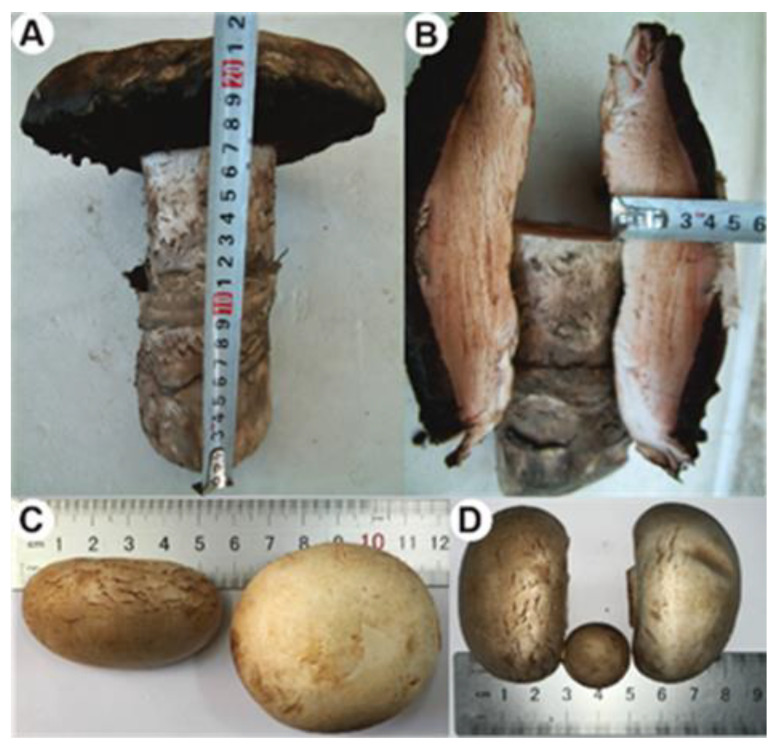
Morphologic comparison in size of the genus *Agaricus* mushrooms. (**A**,**B**) The fruiting bodies of wild *A. bitorquis* BH01. (**C**,**D**) The fruiting bodies of cultivated *A. bisporus* var. *bisporus* H97.

**Figure 2 jof-09-00485-f002:**
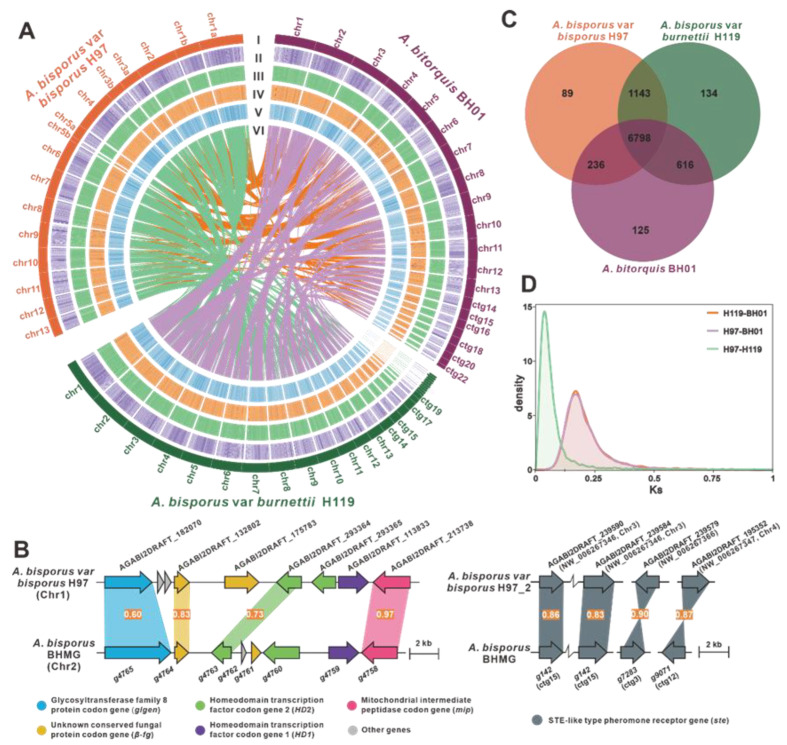
Genomic characterization, mating-type loci, and comparative genomic analysis. (**A**) Genomic collinearity analysis between *A. bitorquis* and *A. bisporus* species. From the outside to the inside are I. Chromosome and Contigs; II–IV. GC-density, GC-skew, AT-skew (window size 10 kb), V. Gene-density (window size 100 kb), VI. Whole-genome collinearity analysis based on protein-coding genes between *A. bitorquis* and *A. bisporus* species. (**B**) Structural diagram of the genes on the *matA* locus and *matB* locus of *A. bitorquis*, the numbers on the similarity diagrams indicate the identity between corresponding genes. (**C**) Venn schematic of comparative genomes between *A. bitorquis* and *A. bisporus* species. (**D**) Ks comparison between *A. bitorquis* and *A. bisporus* species.

**Figure 3 jof-09-00485-f003:**
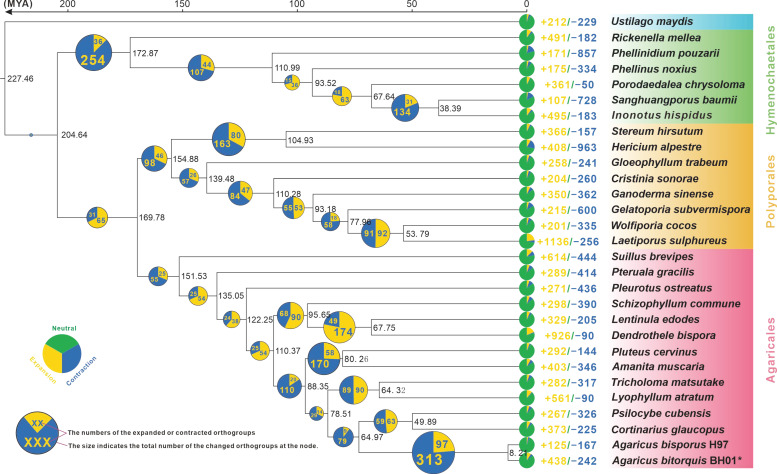
The evolutionary relationship and expanded and contracted gene families among *Agaricus* species and 27 representative Basidiomycetes. The maximum likelihood method credibility tree was inferred from 27 single-copy orthologous genes. All nodes received full bootstrap support. The divergence time is labeled as the mean crown age for each node, while the 95% highest posterior density is also given within the *Agaricus* clade. The black numbers at the branches indicate the corresponding divergence times in millions of years (MYA). The proportion of expansion and contraction in the genome of each species was displayed before its species name.

**Figure 4 jof-09-00485-f004:**
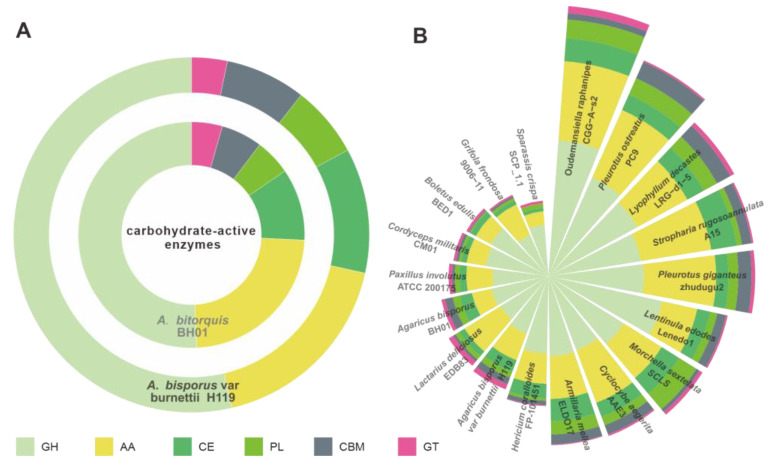
CAZymes analysis of *Agaricus* and related edible mushrooms. (**A**) Composition comparison of CAZymes between *A. bitorquis* BH01 and *A. bisporus* var. burnettii H119. (**B**) Composition comparison of CAZymes between *Agaricus* species and related edible mushrooms.

**Figure 5 jof-09-00485-f005:**
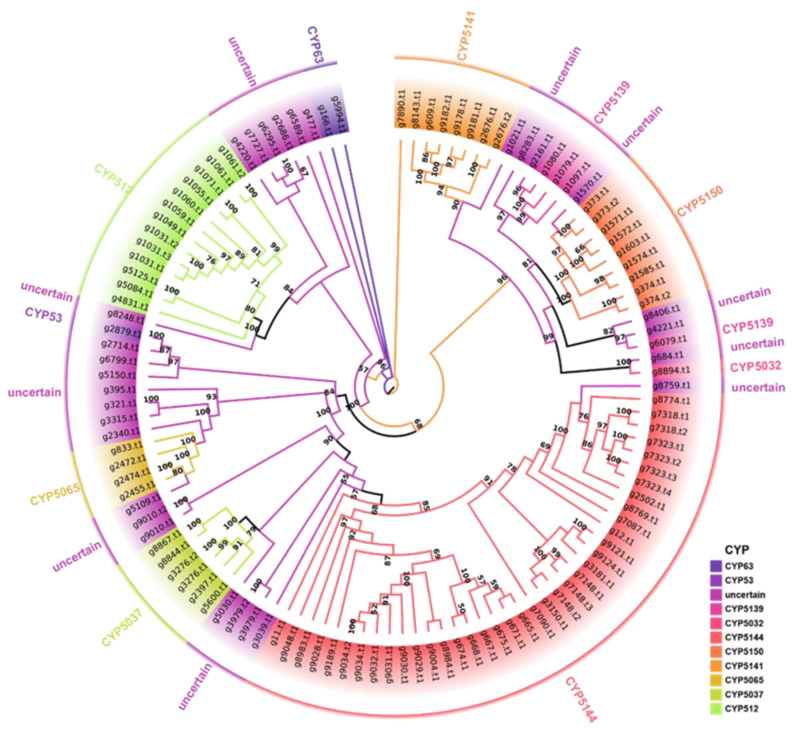
Maximum likelihood tree of 115 cytochrome P450s from *A. bitorquis* BH01. Each cytochrome P450 family is shown in a separate color, and the branch reliability value of over than 50 is marked on the corresponding branch node.

**Figure 6 jof-09-00485-f006:**
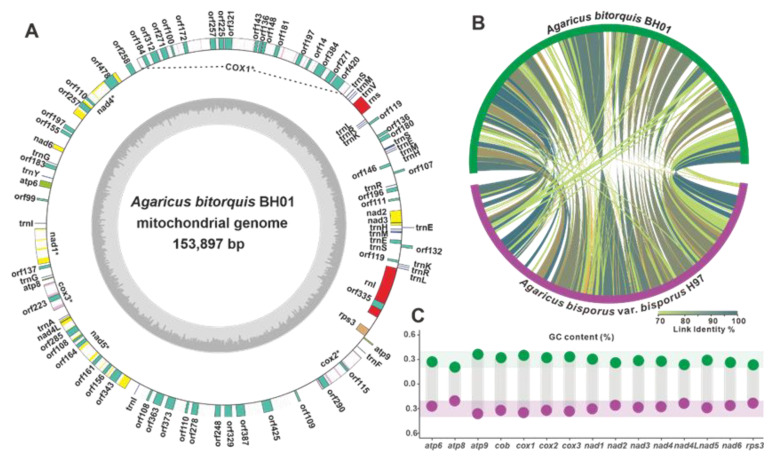
Mitogenome of *A. bitorquis* BH01 and its comparative analyses. (**A**) Circular map of *A. bitorquis* BH01 mitogenome. (**B**) Genomic collinearity analysis. (**C**) GC content comparison of 15 PCGs between *A. bitorquis* BH01 and *A. bisporus* var. bisporus H97.

**Figure 7 jof-09-00485-f007:**
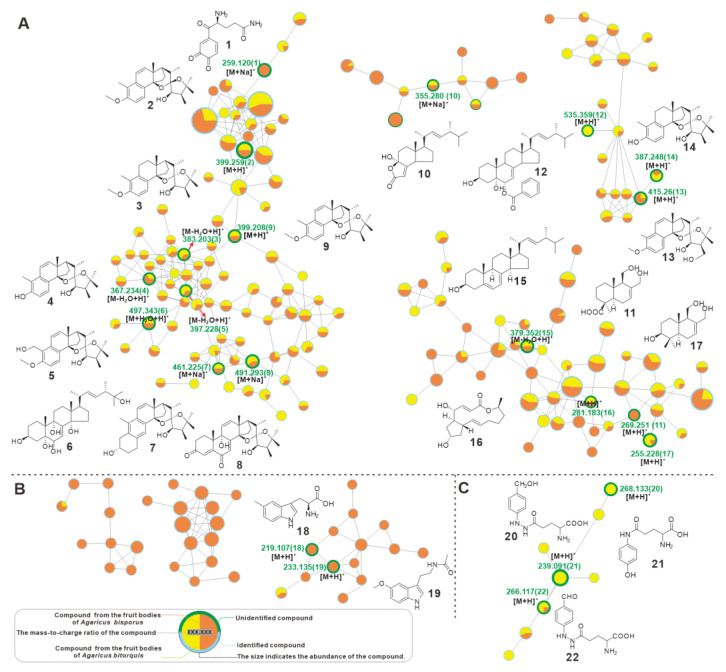
GNPS-based molecular network identification of metabolites from the fruiting bodies of *A. bitorquis* and *A. bisporus*. (**A**) The network of shared molecules. (**B**) The molecular networks unique to *A. bisporus*. (**C**) The molecular network unique to *A. bitorquis*.

**Table 1 jof-09-00485-t001:** Genomic comparison within *Agaricus* species.

Entry	*A. bitorquis* BH01	*A. bisporus* var. *bisporus* H97	*A. bisporus* var. *burnettii* H119
Sequencing technology	lumina NovaSeq 6000 Nanopore PromethION	Sanger dideoxy	PacBio RSII
Sequencing depth	145×	8.29×	70×
No. of contigs	22	70	16
Total length (bp)	32,345,193	30,417,844	30,702,502
Largest length (bp)	3,210,751	3,550,205	3,639,502
Scaffold N50 (bp)	2,052,213	2,550,681	1,931,181
Scaffold L50	10	5	6
GC content (%)	46.05	46.5	46.6
No. of protein-coding genes	10,028	10,438	10,421
GenBank accession No.	PRJNA946023	GCA_000300575.1	GCA_014872705.1
References	This study	[23]	[25]

**Table 3 jof-09-00485-t003:** The identified metabolites from *A. bitorquis* and/or *A. bisporus*.

No	Putative Metabolite	Molecular Formula	Adduct	m/z	Source		Reference
					*A. bisporus*	*A. bitorquis*	
1	γ-L-Glutaminyl-3,4-benzoquinone	C_11_H_12_N_2_O_4_	[M + Na]^+^	259.12	√	NA	Weaver, et al. [38]
2	blazeispirol B	C_25_H_34_O_4_	[M + H]^+^	399.259	√	√	Masao, et al. [39]
3	blazeispirol C	C_25_H_36_O_4_	[M-H_2_O + H]^+^	383.203	√	√	Masao, et al. [39]
4	blazeispirol D	C_24_H_32_O_4_	[M-H_2_O + H]^+^	367.234	√	√	Masao, et al. [39]
5	blazeispirol E	C_25_H_34_O_5_	[M-H_2_O + H]^+^	397.228	√	√	Masao, et al. [39]
6	(22*E*,24*R*)-3*β*,5*α*,6*β*,9*α*,14*α*,25-hexhydroxyergosta-7,22-diene	C_28_H_46_O_6_	[M + H_2_O + H]^+^	497.343	√	√	Rao, et al. [40]
7	blazeispirol X	C_28_H_38_O_4_	[M + Na]^+^	461.225	√	√	Masao, et al. [39]
8	blazeispirol Y	C_28_H_36_O_6_	[M + Na]^+^	491.293	√	√	Masao, et al. [39]
9	blazeispirol A	C_25_H_34_O_4_	[M + H]^+^	399.208	√	√	Masao, et al. [39]
10	demethyl-incisterol A_3_	C_21_H_32_O_3_	[M + Na]^+^	355.28	√	√	Yumi, et al. [41]
11	11,12-dihydroxy-15-drimeneoic acid	C_15_H_24_O_4_	[M + H]^+^	269.251	√	NA	Zhao, et al. [42]
12	benzoyl ergostane	C_35_H_50_O_4_	[M + H]^+^	535.359	NA	√	Yumi, et al. [41]
13	blazeispirol I	C_25_H_34_O_5_	[M + H]^+^	415.26	√	√	Masao, et al. [43]
14	blazeispirol F	C_24_H_35_O_4_	[M + H]^+^	387.248	√	√	Masao, et al. [43]
15	ergosterol	C_28_H_44_O	[M-H_2_O + H]^+^	379.352	√	√	Takeshi, et al. [44]
16	brefeldin A	C_16_H_24_O_4_	[M + H]^+^	281.183	√	√	Dong, et al. [45]
17	3*β*,11,12-trihydroxydrimene	C_15_H_26_O_3_	[M + H]^+^	255.228	√	√	Zhao, et al. [42]
18	5-CH_3_-Tryptophan 5-methyltryptamine	C_12_H_14_N_2_O_2_	[M + H]^+^	219.107	√	NA	Muszyńska, et al. [46]
19	melatonin	C_13_H_16_N_2_O_2_	[M + H]^+^	233.135	√	NA	Muszyńska, et al. [46]
20	agaritine	C_12_H_17_N_3_O_4_	[M + H]^+^	268.133	NA	√	Levenberg
21	N-(γ-L-glutamyl)-4-hydroxyaniline	C_11_H_14_N_2_O_4_	[M + H]^+^	239.091	NA	√	Tsuji, et al. [47]
22	β-*N*-(γ-glutamyl)-4-formylphenyl-hydrazine	C_12_H_15_N_3_O_4_	[M + H]^+^	266.117	NA	√	Albert J., et al. [48]

NA indicates not available.

## Data Availability

The network file based on positive-ion mode MS data can be found and accessed at https://gnps.ucsd.edu/ProteoSAFe/status.jsp?task=6e5a34d20bc14673a01f9e61627f73a5 (accessed on 17 March 2023).

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
