# Peer review of "The First Whole Genome Sequencing of Agaricus bitorquis and Its Metabolite Profiling"

_jof, 2023, doi:10.3390/jof9040485_

Round 1

Reviewer 1 Report

The manuscript is well organized, thought of and presented. This is a high quality work and it deserves to be published. Personally, I absolutely agree with the authors that A. bitorquis is a very interesting species with much higher quality of fruitbodies than A. bisporus and the introduction of this species to the successful growing protocol is highly potential field. However, the species has been detected in much wider territory then cited here, and authors mix taxons: "variety" is not the same as "species". Please, add DNA extraction protocol or cite the source. After these small changes, in my opinion, manuscript will be ready for publication. Please, find enclosed PDF with comments.

Author Response

Opinion: The manuscript is well organized, thought of and presented. This is a high quality work and it deserves to be published. Personally, I absolutely agree with the authors that A. bitorquis is a very interesting species with much higher quality of fruitbodies than A. bisporus and the introduction of this species to the successful growing protocol is highly potential field. However, the species has been detected in much wider territory then cited here, and authors mix taxons: "variety" is not the same as "species". Please, add DNA extraction protocol or cite the source. After these small changes, in my opinion, manuscript will be ready for publication. Please, find enclosed PDF with comments.

Response: Thank you for your generous appreciation of the article. Your professional advice has been very significantly helpful to us. Following your advice, we have cited the detailed protocol of DNA extraction and modified the inappropriate or ambiguous statements in the manuscript. The following are the answers to your doubts

Q1: Line 44: Please cite more reliable data, for example https://www.gbif.org/species/5243374

A1: Thanks for your careful review. We have corrected the ambiguous sentence and cite the website.

Q2: Line 47: ???? is this reliable information? How do you detect them? Personally, I have always detected them as a usual fruiting bodies a bit barren, but 50cm??

A2: We are sorry about this carelessness. The mycorrhizal portion of A. bitorquis is approximately 15cm instead of 50cm. The length of the mycorrhizal portion is also roughly reflected in Figure 1.

Q3: Line66: And many others - please restructure the sentence to be accurate.

A3: Thanks for your advice. We have corrected this sentence in manuscript.

Q4: Line 87: Maybe mycelia? You cannot cultivate fruiting bodies in 3-4 days.

A4: Thanks for pointing out this mistake. That should be mycelia instead of fruiting body. We cut the surface-sterilized fruiting body into pieces and then put them on PDA plates to get mycelia. We have corrected this sentence to offer a better reading experience.

Q5: Line 99: Please, either provide citation or describe the protocol.

A5: Thanks for your advice. We would cite the related citation.

Q6: Lin 377: Not species, but variants of A. bisporus

A6: Thanks for your professional advice. We have corrected them in manuscript.

Reviewer 2 Report

All Latin names of mushrooms should be written in italics. Therefore, please make changes on lines 84, 140, 154, 155, 158, 166, 181, 379.

Author Response

Opinion: All Latin names of mushrooms should be written in italics. Therefore, please make changes on lines 84, 140, 154, 155, 158, 166, 181, 379.

Response: Thanks for your careful review. Your professional suggestion is very important to us. All the fungus names related formatting issues have been corrected.

Reviewer 3 Report

The researchers sequenced the nuclear and mitochondrial genome of Agaricus bitorquis, relative to A. bisporus. Also, they did metabolite profiling. Overall, the genome analysis is basic and more like a genome report. Although, metabolite profiling can be useful for other researchers. Overall analysis quality and writing are good, too. The figures can be improved a lot.

Title:

The title is too long. "The First Whole Genome Sequencing of Agaricus bitorquis" already covers comparative analysis, genome, and mitogenome. Rather, you could include what's special about Agaricus bitorquis, such as (sister to A. bisporus or wild edible mushroom) to get attention from audience. The word "metabolite profiling" looks good to be included in the title.

Lines 14-16:

the whole genome of mitochondrial genome (mitogenome) of → and (typo?)

Line 39, Line 42, and so on

No space at "century[1,2]", it should be "century [1,2]".

→ I see other occurrences of this lack of spaces. Please fix all.

Lines 84, 226, 140, 154, 155, 158, 161, 166, 181, 406, and possibly more:

Please make the species name italics

Lines 87-89:

"The identified mycelium was then deposited in the Shaanxi Key Laboratory of Natural Products & Chemical Biology, College of Chemistry & Pharmacy, Northwest A&F University. " → do you have any accession number so other researchers can get the samples?

Lines 100-102:

"These procedures were carried out meticulously to ensure optimal quality of the extracted DNA for further downstream molecular applications." → I am not sure if this sentence is necessary.

Lines 119-121:

"The rigorous execution of these procedures ensured the accuracy and reliability of the final genome assembly." → I am not sure if this sentence is necessary.

Line 122:

"BRAKER v2.1.4 (https://github.com/Gaius-Augustus/BRAKER) was primarily used" → I wonder if what options of BRAKER you used. Did you use transcriptome data to run the BRAKER?

Lines 133-134:

"These methods were essential to ensure the accuracy and completeness of the genomic 133 annotation." → I am not sure if this sentence is necessary.

Lines 136-137:

https://opensourcelibs.com/lib/mcscanx → it wasn't accessible for me. Could you double-check if this address is correct?

Line 143:

"ParaAT" → could you put version of the software you used?

Line 173:

"XXX-fold" → please fill the number

Line 203:

"Twenty grams of fresh substrates of A and B" → I'm not familiar with metabolite profiling. What are A and B?

Line 212:

"2.8. Data availability" → There's a specific section (line 544) for Data availability. You may move the section there? Also, the genome data under PRJNA946023 isn't available yet. I hope it's in progress. Please don't forget uploading the annotation as well because you referred gene names such as g142 in the manuscript.

Line 228:

BUSCO value of 90.8% → which OrthoDB database did you use? (odb10_eukaryota? odb10_fungi? something else?)

→ Also, BUSCO 90.8% for a chromosome level genome is really low. You may double-check your assembly method. For example, you can use only Illumina DNA-seq reads to assemble and check the BUSCO again?

Line 230:

What is BAUGUSUTS? is it typo for Augustus?

Line 242:

Nr → "NCBI nr" would be better name

Table 1:

No. of scaffolds and contigs → when we usually talk about contigs, they include number of contigs within scaffolds. So the number of contigs should be greater than or equal to the number of scaffolds like H97 and H119. BH01 has 13 scaffolds and 9 contigs, which doesn't make much sense. 

Table 1:

Please include BUSCO completeness, too

Line 257:

"with the top 20 motifs representing the highest number of genes" → it was hard to understand what it means

Line 280:

"indicating the presence of a tetrapolar mating system" → is there any reference to support this? (MAT-A and MAT-B being in the different contigs suggest tetrapolar mating system)

Figure 3:

Why is there only one A. bisprous although you analyzed H97 and H119?

Figure 4A:

Barchart might be better

Figure 4B:

I am not sure it's necessary. You may move it to supplementary or delete. It's hard to interpret (such as comparing between species).

Lines 351-352:

"Notably, g7323 had four transcripts, while g1031 and g7148 each had three transcripts." → I guess when you say "four transcripts" or "three transcripts", that means isoforms? If you predicted that using Braker, that's alternative gene prediction, rather than true isoforms. As long as you don't have strong evidence of isoform (such as transcriptome support), I would that that part out. That is because isoforms in fungi are really rare.

Lines 367-369:

"the genome assembly quality of A. bitorquis BH01 is superior..." → I strongly disagree with that more continuity indicates better assembly quality. That is because it doesn't reflect the accuracy of the assembly. BUSCO completeness could be the indicator how complete they are.

Lines 381-383

"This unexpected difference suggested that the intraspecific differences of the two most common A. bisporus species were greater than the interspecific differences between them and A. bitorquis." → You may draw a phylogenetic tree between three species. It maybe just mislabeling of species name.

Figure 7:

Because there are so many nodes and edges, it's hard to interpret the figure. Can you make it simpler? I would drop the network and list common and unique molecules.

Author Response

Opinion: The researchers sequenced the nuclear and mitochondrial genome of Agaricus bitorquis, relative to A. bisporus. Also, they did metabolite profiling. Overall, the genome analysis is basic and more like a genome report. Although, metabolite profiling can be useful for other researchers. Overall analysis quality and writing are good, too. The figures can be improved a lot.  

Response: Thank you for your careful review and so many valuable advice. Your responsibility as a reviewer has been fully demonstrated. Your detailed professional suggestion is very important to us. We would improve our manuscript as you advised.

Title:

The title is too long. "The First Whole Genome Sequencing of Agaricus bitorquis" already covers comparative analysis, genome, and mitogenome. Rather, you could include what's special about Agaricus bitorquis, such as (sister to A. bisporus or wild edible mushroom) to get attention from audience. The word "metabolite profiling" looks good to be included in the title.

Response: Thank you very much for your in-depth understanding and precise grasp of the content of the manuscript. Following your advice, the current title has been changed to "The First Whole Genome Sequencing of Agaricus bitorquis and its Metabolite Profiling". This title is more concise and informative.

Lines 14-16:

the whole genome of mitochondrial genome (mitogenome) of → and (typo?)

 Response: Thanks for your inspection. We have corrected this mistake.

Line 39, Line 42, and so on

No space at "century[1,2]", it should be "century [1,2]".

→ I see other occurrences of this lack of spaces. Please fix all.

 Response: Thanks for your careful review. We have fixed the related errors.

Lines 84, 226, 140, 154, 155, 158, 161, 166, 181, 406, and possibly more:

Please make the species name italics

 Response: Thanks for your advice. All the fungus names related formatting issues have been corrected.

Lines 87-89:

"The identified mycelium was then deposited in the Shaanxi Key Laboratory of Natural Products & Chemical Biology, College of Chemistry & Pharmacy, Northwest A&F University. " → do you have any accession number so other researchers can get the samples?

 Response: Everyone is welcome to contact the author via email to obtain the related data or strain for free.

Lines 100-102:

"These procedures were carried out meticulously to ensure optimal quality of the extracted DNA for further downstream molecular applications." → I am not sure if this sentence is necessary.

 Response: Thanks for your advice. We have made some changes in manuscript to offer a better reading experience.

Lines 119-121:

"The rigorous execution of these procedures ensured the accuracy and reliability of the final genome assembly." → I am not sure if this sentence is necessary.

 Response: Thanks for your advice. We have made some changes in manuscript to offer a better reading experience.

Line 122:

"BRAKER v2.1.4 (https://github.com/Gaius-Augustus/BRAKER) was primarily used" → I wonder if what options of BRAKER you used. Did you use transcriptome data to run the BRAKER?

 Response: We performed transcriptome sequencing in parallel with genome sequencing. Transcriptome data were used to assist in annotating the genome. In addition, there are de novo and homologous protein-based methods for annotating genomes using BRAKER.

Lines 133-134:

"These methods were essential to ensure the accuracy and completeness of the genomic 133 annotation." → I am not sure if this sentence is necessary.

 Response: Thanks for your advice. We have made some changes in manuscript to offer a better reading experience.

Lines 136-137:

https://opensourcelibs.com/lib/mcscanx → it wasn't accessible for me. Could you double-check if this address is correct?

 Response: We are sorry about the invalid website. The new website has replaced the invalid one.

Line 143:

"ParaAT" → could you put version of the software you used?

 Response: We have attached the version information of ParaAT and other related software mentioned in manuscript.

Line 173:

"XXX-fold" → please fill the number

 Response: We are sorry about this carelessness. The average genome coverage is 150. We have corrected the sentence.

Line 203:

"Twenty grams of fresh substrates of A and B" → I'm not familiar with metabolite profiling. What are A and B?

 Response: We are sorry about this carelessness. A & B should be “A. bitorquis” & “A. bisporus”.

Line 212:

"2.8. Data availability" → There's a specific section (line 544) for Data availability. You may move the section there? Also, the genome data under PRJNA946023 isn't available yet. I hope it's in progress. Please don't forget uploading the annotation as well because you referred gene names such as g142 in the manuscript.

 Response: Thanks for your advice. We have moved that to the section “Data availability”. We have uploaded the genome and annotation file. And the submission seems still under processing. We would further add the annotation information in supplementary material.

Line 228:

BUSCO value of 90.8% → which OrthoDB database did you use? (odb10_eukaryota? odb10_fungi? something else?)

→ Also, BUSCO 90.8% for a chromosome level genome is really low. You may double-check your assembly method. For example, you can use only Illumina DNA-seq reads to assemble and check the BUSCO again?

 Response: We used the 'odb10_fungi' database and added this information where appropriate in the revised manuscript.

In addition, we have revised the relevant statements about BUSCO to make them more rigorous and scientific. Once again, we thank the reviewers for their careful attention to this detail.

Line 230:

What is BAUGUSUTS? is it typo for Augustus?

 Response: Thank you for your careful review. We have corrected this typo.

Line 242:

Nr → "NCBI nr" would be better name

 Response: Thanks for your advice. We have corrected it.

Table 1:

No. of scaffolds and contigs → when we usually talk about contigs, they include number of contigs within scaffolds. So the number of contigs should be greater than or equal to the number of scaffolds like H97 and H119. BH01 has 13 scaffolds and 9 contigs, which doesn't make much sense. 

 Response: Thanks for your detailed advice. We would correct this misconception and modify the manuscript.

Table 1:

Please include BUSCO completeness, too

 Response: Thanks for your constructive suggestion. In fact, we also wanted to compare this parameter at first, but BUSCO for H97 is not available. In addition, BUSCO for H119 is not clear, it is only described as greater than 99%.

Line 257:

"with the top 20 motifs representing the highest number of genes" → it was hard to understand what it means

 Response: This sentence has been revised to be more readable and understandable.

Line 280:

"indicating the presence of a tetrapolar mating system" → is there any reference to support this? (MAT-A and MAT-B being in the different contigs suggest tetrapolar mating system)

 Response: Thank you for your careful consideration, and the additional citations will make this statement more justifiable. Corresponding literature has been added.

Figure 3:

Why is there only one A. bisprous although you analyzed H97 and H119?

 Response: We aim to illustrate phylogenetic relationship of A. bisprous and A. bitorquis in Basidiomycetes family. Thus, we just choose one variant of A. bisprous to conduct phylogenetic tree. The corresponding markers have been added.

Figure 4A:

Barchart might be better

 Response: Thanks for your kindly advise. This is a circle form of bar chart. To match the style of Figure 4B.

Figure 4B:

I am not sure it's necessary. You may move it to supplementary or delete. It's hard to interpret (such as comparing between species).

 Response: Figure 4B illustrates composition of CAZymes between Agaricus species and related edible mushrooms. To provide a concise and clear figure, we make a slight change in Figure 4.

Lines 351-352:

"Notably, g7323 had four transcripts, while g1031 and g7148 each had three transcripts." → I guess when you say "four transcripts" or "three transcripts", that means isoforms? If you predicted that using Braker, that's alternative gene prediction, rather than true isoforms. As long as you don't have strong evidence of isoform (such as transcriptome support), I would that that part out. That is because isoforms in fungi are really rare.

 Response: Your analysis makes sense; we really don't know whether to use "transcripts" or "isoforms". The transcriptome data used to assist in annotating the genome is obtained by next-generation sequencing, not three-generation full-length transcriptome. To avoid this possible misleading information, this detailed description has been deleted. This deletion will not have any impact on the manuscript. Thanks again to the reviewer for his/her meticulousness and rigor.

Lines 367-369:

"the genome assembly quality of A. bitorquis BH01 is superior..." → I strongly disagree with that more continuity indicates better assembly quality. That is because it doesn't reflect the accuracy of the assembly. BUSCO completeness could be the indicator how complete they are.

 Response: We strongly agree with you that the quality of the assembly cannot be reflected by only some of the parameters. At the same time, this statement has been removed, considering that the assembly completeness is only 90.8%. This treatment makes the manuscript more objective and rigorous.

Lines 381-383

"This unexpected difference suggested that the intraspecific differences of the two most common A. bisporus species were greater than the interspecific differences between them and A. bitorquis." → You may draw a phylogenetic tree between three species. It maybe just mislabeling of species name.

 Response: Thanks for you careful revie. We did mislabel the species names in this sentence. We reconduct the phylogenetic tree among these three species and the two A. bisporus variants still show closest relation. And two variant of A. bisporus did share more homologous proteins. We have corrected this sentence.

Figure 7:

Because there are so many nodes and edges, it's hard to interpret the figure. Can you make it simpler? I would drop the network and list common and unique molecules.

Response: Thanks for your advice. We have deleted some irrelevant information to illustrate their metabolite profiling as succinct as possible.